# Revisiting Model Inversion Evaluation: From Misleading Standards to Reliable Privacy Assessment

## Abstract

Model Inversion (MI) attacks aim to reconstruct information from private training data by exploiting access to machine learning models $T$. To evaluate such attacks, the standard evaluation framework relies on an evaluation model $E$, trained under the same task design as $T$. This framework has become the de facto standard for assessing progress in MI research, used across nearly all recent MI attacks and defenses without question. **In this paper,** we present the first in-depth study of this MI evaluation framework. In particular, we identify a critical issue of this standard MI evaluation framework: *Type-I adversarial examples*. These are reconstructions that do not capture the visual features of private training data, yet are still deemed successful by the target model $T$ and ultimately transferable to $E$. Such false positives undermine the reliability of the standard MI evaluation framework. To address this issue, we introduce a new evaluation framework, $\mathcal{F}_{MLLM}$, which replaces $E$ with advanced Multimodal Large Language Models (MLLMs). We propose systematic design principles for $\mathcal{F}_{MLLM}$. By leveraging their general-purpose visual understanding, our MLLM-based framework does not depend on training of shared task design as in $T$, thus reducing Type-I transferability and providing more faithful assessments of reconstruction success. Using our proposed evaluation framework, we reevaluate 27 diverse MI attack setups and empirically reveal consistently high false positive rates under the standard evaluation framework. Importantly, we demonstrate that many state-of-the-art (SOTA) MI methods report inflated attack accuracy, indicating that actual privacy leakage is significantly lower than previously believed. By uncovering this critical issue and proposing a robust solution, our work enables a reassessment of progress in MI research and sets a new standard for reliable and robust evaluation. **Our MLLM-based MI evaluation framework and benchmarking suite are included in the Appendix.**

## 1 Introduction

Model Inversion (MI) attacks pose a significant privacy threat by attempting to reconstruct confidential information from sensitive training data through exploiting access to machine learning models. Recent state-of-the-art (SOTA) MI attacks (Zhang et al., 2020; Chen et al., 2021; Wang et al., 2021a; Nguyen et al., 2023a;b; Qiu et al., 2024; Yuan et al., 2023) have shown considerable advancements, reporting attack success rates exceeding 90%. This vulnerability is particularly alarming for security-sensitive applications such as face recognition (Meng et al., 2021; Guo et al., 2020; Huang et al., 2020; Schroff et al., 2015), medical diagnosis (Dufumier et al., 2021; Yang et al., 2022; Dippel et al., 2021), or speech recognition (Chang et al., 2020; Krishna et al., 2019).

**Research gap.** Recently, there are many studies on improving MI attacks (Zhang et al., 2020; Fredrikson et al., 2014; An et al., 2022; Chen et al., 2021; Yuan et al., 2023; Qiu et al., 2024; Nguyen et al., 2023b;a; Kahla et al., 2022; Han et al., 2023) and MI defenses (Wang et al., 2021b; Peng et al., 2022; Struppek et al., 2024b; Ho et al., 2024; Koh et al., 2024). To assess the effectiveness of these MI attacks/defenses, MI Attack Accuracy (AttAcc) is a standard and the most important metric. To measure AttAcc, almost all recent MI studies adopt the evaluation framework introduced by (Zhang et al., 2020), which we denote as $\mathcal{F}_{Curr}$. Under $\mathcal{F}_{Curr}$, an evaluation model $E$ is used to predict the identities of individuals based on the MI-reconstructed images. Model $E$ is trained on the same

**MI Results**    **Private Training Data**

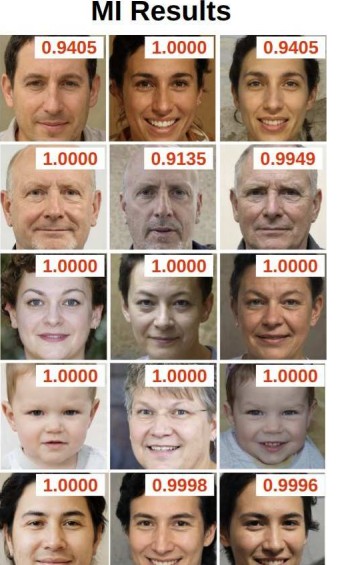

Figure 1: **In this work, we present the first and in-depth study on the Model Inversion (MI) evaluation**. Particularly, we investigate the most common MI evaluation framework $\mathcal{F}_{Curr}$ to measure MI Attack Accuracy (AttAcc). $\mathcal{F}_{Curr}$ is introduced in (Zhang et al., 2020) and is utilized to assess almost all recent MI attacks/defenses. However, we find that $\mathcal{F}_{Curr}$ suffers from a significant number of false positives. These false positive MI reconstructed samples do not capture visual identity features of the target individual in the private training data, but they are still deemed successful attacks according to $\mathcal{F}_{Curr}$ with a high confidence (indicated in red text). Extensive visualization of false positives are included in the Appx. C

private dataset and follows the same training task design as in target model $T$. In particular, consider the scenario where the adversary targets a specific identity $y$. From this point onward, we use $y$ exclusively to denote the target class (label) in the MI attack, rather than a label predicted by any model. The adversary produces reconstructed images $x_y^r$ of the target label $y$ by exploiting access to the target model $T$. To measure AttAcc, these $x_y^r$ samples are then passed through an evaluation model $E$. According to $\mathcal{F}_{Curr}$, the attack is deemed successful if $E$ classifies $x_y^r$ as $y$. **Even though such MI evaluation framework becomes the de facto standard and the accuracy measured by $\mathcal{F}_{Curr}$ has been the most important metric in gauging MI research progress in almost all recent MI studies, there has not been any in-depth and comprehensive study to understand the accuracy of $\mathcal{F}_{Curr}$ and its limitations.**

**In this work,** we conduct the first in-depth study of the standard MI evaluation framework $\mathcal{F}_{Curr}$. For a truly successful attack, the reconstructed images $x_y^r$ should capture the visual identity features of $y$. However, we find that there exists a considerable number of MI reconstructed images that lack visual identity features of $y$, yet both target model $T$ and evaluation model $E$ under $\mathcal{F}_{Curr}$ still assign high probabilities to $y$, i.e., high values of $P_E(y|x_y^r)$. Some examples are illustrated in Fig. 1. **These *false positives* potentially inflate the reported success rate of recent SOTA MI attacks.**

To shed light on the causes of these false positives, we systematically discover the impact of Type I Adversarial Attacks (Nguyen et al., 2015; Tang et al., 2019) in Model Inversion, highlighting a connection between two previously distinct research areas. The optimization processes in MI attacks and Type I adversarial attacks are similar: both maximizing the likelihood with respect to (w.r.t.) the input under a fixed model. We show that false positives in MI and Type I adversarial examples are mathematically equivalent: **MI false positives and Type I adversarial examples are essentially the same construct mathematically, only arising under different problem contexts, MI versus adversarial attacks.** Moreover, due to the well-documented phenomenon of *adversarial transferability* (Nguyen et al., 2015), these false positives can transfer to the MI evaluation model $E$. This transferability is especially pronounced when the evaluation model $E$ in $\mathcal{F}_{Curr}$ shares the same task design as the target model $T$ (Liang et al., 2021; Papernot et al., 2016a). Ultimately, we demonstrate a fundamental issue in $\mathcal{F}_{Curr}$, potentially leading to unreliable assessment and inflated success rates reported for recent MI attacks under this evaluation framework.

To mitigate this issue, we propose a new MI evaluation framework $\mathcal{F}_{MLLM}$ that replaces the evaluation model $E$ with Multimodal Large Language Models (MLLMs). We propose systematic design principles for $\mathcal{F}_{MLLM}$. While not relying on the same training task design used in $T$, $\mathcal{F}_{MLLM}$ minimizes Type-I transferability and offers a more faithful assessment of MI attacks. Using data annotated by $\mathcal{F}_{MLLM}$, we empirically benchmark the reliability of the common MI evaluation framework $\mathcal{F}_{Curr}$ and reveal consistently high false positive rates across 27 diverse MI setups. Our findings challenge the reliability of the standard MI evaluation framework and underscore the importance of adopting our MLLM-based approach for more reliable assessments. Our main contributions are summarized below:

- We present the first in-depth study on the most common evaluation framework $\mathcal{F}_{Curr}$ to compute MI AttAcc. Our study identify the surprising effect of Type I Adversarial Features, as well as adversarial transferability (Nguyen et al., 2015; Tang et al., 2019), on MI attacks, highlighting a relationship between two previously distinct research areas (see Sec. 3). Our findings explain numerous false positives in $\mathcal{F}_{Curr}$.

- To mitigate this issue, we propose an MLLM-based MI evaluation framework $\mathcal{F}_{MLLM}$ (see Sec. 4.1). Our framework leverages the powerful general-purpose visual understanding capabilities of advanced MLLMs to evaluate MI reconstructions, replacing the traditional evaluation model $E$. We propose systematic design principles for $\mathcal{F}_{MLLM}$. Without relying on the same training task design used in $T$, $\mathcal{F}_{MLLM}$ minimizes Type-I transferability and provides a more faithful assessment of MI attacks.

- Using data annotated by our evaluation framework, we empirically demonstrate that there are number type I adversarial examples under $\mathcal{F}_{Curr}$. Ultimately, our findings challenge the validity of this dominant evaluation framework and underscore the importance of adopting our proposed MI evaluation framework for more reliable assessments (see Sec. 4.2).

## 2 RELATED WORK

Model Inversion (MI) aims to extract information about the training data given a trained model. Particularly, an adversary exploits a target model $T$ that was trained on a private dataset $\mathcal{D}_{priv}$. However, $\mathcal{D}_{priv}$ should not be disclosed. The main goal of MI attacks is to extract information about the private samples in $\mathcal{D}_{priv}$. The existing literature formulates MI attacks as a process of reconstructing an input $x_y^r$ that $T$ is likely to classify into the target class (label) $y$. For example, in facial recognition, MI attacks aim to reconstruct facial images that are likely to be identified as belonging to a particular person.

**Model Inversion Attacks.** One of the initial methods for MI is proposed by Fredrikson et al. (Fredrikson et al., 2014), who discover that attackers could use a machine learning model to extract genomic and demographic information about patients. Their work is extended to facial recognition (Fredrikson et al., 2015), demonstrating the potential to reconstruct identifiable facial images from model outputs. Advancing this concept, Yang et al. (2019) propose adversarial model inversion, treating the target model as an encoder and using a secondary network to reconstruct the original input data from the prediction vector. Recent advanced generative-based MI attack methods propose reducing the search space to the latent space by training a deep generator (Zhang et al., 2020; Wang et al., 2021a; Chen et al., 2021; Yang et al., 2019; Yuan et al., 2023; Nguyen et al., 2023a; Struppek et al., 2022; Qiu et al., 2024), instead of directly performing MI attacks on high-dimensional space such as the image space. Specifically, GMI (Zhang et al., 2020) and PPA (Struppek et al., 2022) employ pretrained GAN models (e.g., WGAN (Arjovsky et al., 2017) or StyleGAN (Karras et al., 2019)) on an auxiliary dataset similar to private training data $\mathcal{D}_{priv}$. Inversion images are found through the latent vector of the generator. Recent efforts have aimed to enhance GAN-based MI methods from multiple perspectives. From the perspective of prior knowledge, KEDMI (Chen et al., 2021) proposes training inversion-specific GANs using knowledge from the target model $T$. Similarly, Pseudo-Label Guided MI (Yuan et al., 2023) utilizes pseudo-labels to guide conditional GAN training, while IF-GMI (Qiu et al., 2024) leverages intermediate feature representations from pretrained GAN blocks. From MI objective perspective, max-margin loss (Yuan et al., 2023) and logit loss (Nguyen et al., 2023a) are introduced to address limitations in Cross-Entropy loss used in MI attacks. From MI overfitting perspective, LOMMA (Nguyen et al., 2023a) employs augmented model concepts to

improve generalizability of MI attacks. The Eq. 1 represents the general step of SOTA MI attacks:

$$w^* = \arg\min_w(-\log P_T(y|G(w)) + \lambda\mathcal{L}_{prior}(w)) \tag{1}$$

Here, $-\log \mathcal{P}_T(y|G(w))$ represents the identity loss, guiding the reconstruction of $x_y^r = G(w)$ that is most likely to be classified as target class $y$ by target classifier $T$. The $\mathcal{L}_{prior}$ is a prior loss, using public information to establish a distributional prior via GANs, thereby guiding the inversion process towards meaningful reconstructions.

**Model Inversion Defenses.** In contrast to MI attacks, MI defenses aim to minimize the disclosure of training samples during the MI optimization process. To protect against MI attacks, the objective is to create a method to train the target classifier $T$ on $\mathcal{D}_{priv}$ in such a way that $T$ reveals as little information about $\mathcal{D}_{priv}$ as possible, regarding specific labels, while still achieving satisfactory model performance. Efforts have been made to develop defenses against MI attacks. MID (Wang et al., 2021b) adds a regularizer to the target classifier's objective during training, penalizing the mutual information between inputs $x$ and outputs $T(x)$. BiDO (Peng et al., 2022) introduces a bilateral regularizer that minimizes the information about inputs $x$ in feature representations $z$ while maximizing the information about labels $y$ in $z$. Beyond regularization-based defenses, TL-DMI (Ho et al., 2024) improves MI robustness through transfer learning, while LS (Struppek et al., 2024b) employs Negative Label Smoothing to improve MI robustness. Recently, MI robustness is also explored from a architecture perspective (Koh et al., 2024).

**Model Inversion Evaluation Metrics.** To evaluate MI attacks and defenses, almost all existing studies rely on the standard MI evaluation framework from (Zhang et al., 2020), denoted as $\mathcal{F}_{Curr}$, which computes attack accuracy and serves as the main metric for monitoring progress in MI research. Suppose an adversary targets a class $y$ and reconstructs images $x_y^r$ using access to the target model $T$. These $x_y^r$ are then classified by an evaluation model $E$ (trained on the same $\mathcal{D}_{priv}$ but distinct from $T$). Under $\mathcal{F}_{Curr}$, an attack is successful if $E$ predicts $x_y^r$ as $y$. Beyond attack accuracy, several complementary metrics are also used. KNN distance is used in (Zhang et al., 2020; Chen et al., 2021; Nguyen et al., 2023a; Yuan et al., 2023; Struppek et al., 2022) to measure the shortest feature distance between reconstructed and private images of class $y$, using features from $E$ or an external extractor. FID is used in (Peng et al., 2022; Qiu et al., 2024; Struppek et al., 2022; Yuan et al., 2023) to assess the realism of reconstructions. Knowledge Extraction Score is used in (Struppek et al., 2024b) to evaluate discriminative information by training a surrogate classifier on reconstructed images and measuring its accuracy on $T$'s training data. Although these complementary metrics provide additional perspectives, $\mathcal{F}_{Curr}$ remains the dominant evaluation framework and attack accuracy the most widely used measure of progress in MI research. However, despite its prevalence, there has not yet been a comprehensive study examining the reliability and limitations of $\mathcal{F}_{Curr}$. *In this work, we take the first step toward such an investigation.*

## 3 COMMON MI EVALUATION FRAMEWORK HAS ISSUES: A CONNECTION BETWEEN MI ATTACKS AND TYPE I ADVERSARIAL ATTACKS

We discover, for the first time, the strong connection between Type I Adversarial Attacks and MI Attacks. Due to this strong connection, the adversarial type I examples could be generated during MI attacks. Additionally, due to the well-documented phenomenon of adversarial transferability (Nguyen et al., 2015), these adversarial type I examples can transfer to $E$ in $\mathcal{F}_{Curr}$. Ultimately, these phenomenons results in unreliable assessment for $\mathcal{F}_{Curr}$.

### 3.1 AN OVERVIEW OF ADVERSARIAL ATTACKS

An adversarial attack on machine learning models is an intentional manipulation of input data to cause incorrect predictions, highlighting potential vulnerabilities of the model. Adversarial attacks aim to create inputs that deceive machine learning classifiers into making errors while humans do not. There are two main types of adversarial attacks: Type I and Type II. Type II Adversarial Attacks (Goodfellow et al., 2014; Szegedy et al., 2013; Kurakin et al., 2016; Papernot et al., 2016b; Carlini & Wagner, 2017; Moosavi-Dezfooli et al., 2016; Shafahi et al., 2019) are commonly studied and aim to produce false negatives. In this attack, minor and imperceptible perturbations are added to the input data $x$ to generate an adversarial example $x^{advII}$, which is incorrectly classified by the model.

Mathematically, this is represented as:

$$f(x^{\text{adv-II}}) \neq f(x) \text{ and } f_{oracle}(x^{\text{adv-II}}) = f_{oracle}(x) \tag{2}$$

Here, $f$ is the model under attack, $f_{oracle}$ is an oracle or hypothetical, idealized classifier. In addition to Type II attack, Type I Adversarial Attacks are designed in (Nguyen et al., 2015; Tang et al., 2019) to generate false positives by creating examples that are significantly different from the original input but are still classified as the same class by the model. This involves producing an adversarial example $x^{\text{adv-I}}$ that, despite being significantly different from input $x$, the target model $f$ mis-classifies as the original class. **Formally, Type I adversarial attack sample is defined as follows** (Nguyen et al., 2015; Tang et al., 2019):

$$f(x^{\text{adv-I}}) = f(x) \text{ and } f_{oracle}(x^{\text{adv-I}}) \neq f_{oracle}(x) \tag{3}$$

Type I adversarial samples can be produced by optimizing the input by iteratively updating it to maximize the likelihood under a fixed target model classifier (Nguyen et al., 2015). This process optimizes the input to remain within a targeted decision boundary, however, it is different from training data by human perception. This phenomenon can also be viewed as over-confidence phenomenon of machine learning models (Wei et al., 2022; Guo et al., 2017).

## 3.2 THE STRONG CONNECTION BETWEEN MODEL INVERSION ATTACKS AND TYPE I ADVERSARIAL ATTACKS

In the following, we analyze the connection between MI attacks and Type I adversarial attacks. The general inversion step used in SOTA MI attacks is described in Eq. 1. As a result, reconstructed images often exhibit high likelihood under the target classifier $T$. However, not all reconstructed images successfully capture the visual identity features of the target individual from the private training data. Some examples are illustrated in Fig. 1. We refer to these cases as **false positives (under $T$) in MI**. Specifically, with

Table 1: Mathematical equivalence of False Positive (FP) in MI (Eq. 4) and Type I adversarial attack (Eq. 3) (Nguyen et al., 2015; Tang et al., 2019).

| | False Positive in MI | Type I Adversarial Attack |
|---|---|---|
| **Fixed Model Under Attack** | $T$ | $f$ |
| **Private/Original Sample** | $x_y$ | $x$ |
| **Attack Sample** | $x_y^r$ | $x^{\text{adv-I}}$ |
| **Formulation** | $T(x_y^r) = T(x_y)$ | $f(x^{\text{adv-I}}) = f(x)$ |
| | $f_{oracle}(x_y^r) \neq f_{oracle}(x_y)$ | $f_{oracle}(x^{\text{adv-I}}) \neq f_{oracle}(x)$ |

target classifier $T$ and $f_{oracle}$ denoting the oracle, **false positive in MI attack is mathematically represented as**:

$$T(x_y^r) = T(x_y) = y$$
$$\text{and } f_{oracle}(x_y^r) \neq f_{oracle}(x_y) \tag{4}$$

Here, $x_y^r$ does not resemble the visual identity feature of $x_y$. However, $T$ classifies $x_y^r$ as the target label $y$. Hence, $x_y^r$ is a false positive in MI. In Eq. 4, an alternative interpretation is that the false positive $x_y^r$ is the example that can deceive $T$ to classify it as the target label $y$ while oracle classifier can not recognize $x_y^r$ as $y$.

**Critically, by comparing Eq. 4 and Eq. 3, we reveal the mathematical equivalence of MI false positives and Type I adversarial examples: both describe attack samples ($x_y^r$, $x^{\text{adv-I}}$ resp.) optimized under a fixed model ($T$, $f$ resp.) that preserve the model's prediction while deviating from human-perceived identity. This equivalence uncovers a key insight: MI false positives and Type I adversarial examples are essentially the same construct mathematically, only arising under different problem contexts, MI versus adversarial attacks.** Tab. 1 summarizes the equivalence.

## 3.3 ADVERSARIAL TRANSFERABILITY CAN LEAD TO CRITICAL ISSUES IN THE COMMON MODEL INVERSION EVALUATION FRAMEWORK

In $\mathcal{F}_{\text{Curr}}$, an evaluation model $E$ is used to predict the identities of individuals based on MI-reconstructed images. The model $E$ is trained on the same private dataset and follows the same training task design (i.e., an $n$-way classification task). Prior work has shown that adversarial examples crafted for one model can often transfer and mislead other models, even those with different

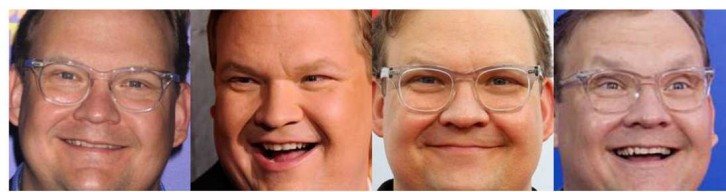

Image A                    Set of Images B

**Does Image A depict the same individual as the images in Set B?**

Figure 2: **An example of evaluation query in our** $\mathcal{F}_{\mathbf{MLLM}}$. The task is to determine whether "Image A" depicts the same individual as those in "Image B". We have two setups: (1) "Image A" and "Image B" consist of private images and (2) "Image A" is an MI-reconstructed image $x_y^r$ of the target label $y$ while four real images of $y$ are randomly selected as "Image B". MLLM is tasked with responding either "Yes" or "No" to indicate whether "Image A" matches the identity in "Image B". The detailed prompt can be found in the Appx. B.2.

architectures, as long as they are trained on the similar dataset and task design (Nguyen et al., 2015; Papernot et al., 2016a; Liang et al., 2021). Therefore, the similarity between $E$ and the target model $T$ may enable the transferability of Type I adversarial examples from $T$ to $E$. This phenomenon can be expressed mathematically as:

$$T(x_y^r) = T(x_y) = y \qquad E(x_y^r) = E(x_y) = y$$
$$\text{and } f_{oracle}(x_y^r) \neq f_{oracle}(x_y) \tag{5}$$

In Sec. 4.3, we provide quantitative results to support our analysis and demonstrate how Type I adversarial perturbations inflate the false-positive rates, leading to overestimation of attack effectiveness.

## 4 OUR PROPOSED MLLM-BASED MODEL INVERSION EVALUATION FRAMEWORK

In this section, we introduce a novel and faithful evaluation framework, $\mathcal{F}_{\mathrm{MLLM}}$. To mitigate the undesirable effects of Type I adversarial transferability under $T$, our key idea is to employ a model trained under a fundamentally different learning regime. We argue that MLLMs are ideal candidates because they are trained on broad, general-purpose tasks using data and optimization pipelines entirely distinct from those of the target model in MI. In fact, MLLMs are increasingly used for automated data labeling, offering robustness and scalability in both academia and industry (Community, 2025; Smith & Chen, 2024; Lee & Patel, 2024; Zhou et al., 2024). However, not every MLLM is suitable for $\mathcal{F}_{\mathrm{MLLM}}$ when evaluating MI problems, even the most SOTA models. To systematically determine an ideal candidate, we propose a principle for designing and selecting MLLMs in Sec. 4.1. Then, we then re-evaluate MI attack accuracy using $\mathcal{F}_{\mathrm{MLLM}}$ in Sec. 4.2. Finally, in Sec. 4.3, we provide quantitative evidence demonstrating the impact of Type I adversarial features on MI.

### 4.1 PRINCIPLE OF DESIGNING AND IMPLEMENTING $\mathcal{F}_{\mathrm{MLLM}}$

$\mathcal{F}_{\mathrm{MLLM}}$ **design.** We outline $\mathcal{F}_{MLLM}$ design in Fig. S.2. Specifically, given a reconstructed image, we form an evaluation query by pairing it with a set of private training images that include the target identity. We then combine this query image with a natural language textual prompt and provide both as input to the MLLM. The prompts shown in Tab. S.6 are fixed across all evaluation queries to ensure fairness. For each reconstructed image, the model outputs a categorical response ("Yes" or "No"), where "Yes" indicates a successful attack. By evaluating many such queries and computing the proportion of correct identifications, $\mathcal{F}_{\mathrm{MLLM}}$ provides an automated and faithful evaluation of MI.

The key lies in selecting a reliable MLLM for faithful MI evaluation. We propose two criteria: **(1) strong capability in understanding interleaved image-text inputs**, and **(2) have no usage restrictions on MI tasks** (e.g., some commercial MLLMs may refuse queries involving human facial data). To quantify these criteria, we test whether the MLLM can accurately recognize samples from a private dataset (Zhang et al., 2020; Struppek et al., 2022). As shown in Fig. 2, we design two test sets: ● Positive pairs: "Image A" is a real image of the same individual present in "Images

Table 2: Following our selection principle, the experiment results indicate that Gemini-2.0 serves as a reliable MLLM for $\mathcal{F}_{\mathrm{MLLM}}$ by demonstrating (1) Strong capability in understandin interleaved image-text inputs and (2) have no usage restrictions on MI tasks.

| MLLM | | "Yes" rate ↑ | "No" rate ↓ | "Refuse" rate ↓ | | "Yes" rate ↓ | "No" rate ↑ | "Refuse" rate ↓ |
|---|---|---|---|---|---|---|---|---|
| Gemini 2.0 | | **93.84%** | **3.16%** | **0%** | | **4.41%** | **95.59%** | **0%** |
| ChatGPT-5 | **Positive Pairs** | 17.50% | 2.67% | 79.83% | **Negative Pairs** | 0.09% | 23.04% | 76.86% |
| Qwen2.5VL | | 88.51% | 11.49% | 0% | | 5.50% | 94.55% | 0% |

B," with the expected answer "Yes." • Negative pairs: "Image A" is a real image of a different individual than those in "Images B," with the expected answer "No". An MLLM is considered a reliable evaluator if it consistently yields high "Yes" rates for Positive pairs and high "No" rates for Negative pairs. Moreover, we expect MLLMs to have a low "Refusal" rate when assessing queries. We conduct this experiment on the widely used MI dataset FaceScrub (Ng & Winkler, 2014). We evaluate several SOTA MLLMs, including ChatGPT-5, Gemini-2.0, and Qwen2.5VL-72B within our $\mathcal{F}_{\mathrm{MLLM}}$ framework.

**Results.** The results are reported in Tab. 2. We observe that Gemini-2.0 achieves high "Yes" rates for Positive pairs and high "No" rates for Negative pairs. Qwen2.5VL-72B may have limited capability of current open-source MLLMs to understand interleaved image-text inputs compared to closed-source commercial models. Despite ChatGPT-5 is a powerful closed-source model, it refuses to assess MI queries with high "Refuse" rates (see examples in Appx. A.1.2). At the end of the day, our principle recommends Gemini-2.0 as a reliable MLLM for $\mathcal{F}_{\mathrm{MLLM}}$. Furthermore, $\mathcal{F}_{\mathrm{MLLM}}$ powered by Gemini-2.0 is robust to randomness, additional MI dataset, and aligns well with human evaluation (See Appx. A.1). Note that while Gemini-2.0 is a strong choice in our study, it is not the only MLLM option for $\mathcal{F}_{\mathrm{MLLM}}$. As MLLMs continue to evolve, we can adopt other models as long as they satisfiy our selection principles.

## 4.2 REASSESSING MI ATTACK ACCURACY USING OUR MLLM-BASED EVALUATION FRAMEWORK

We empirically reassess SOTA MI attacks using our proposed evaluation framework $\mathcal{F}_{\mathrm{MLLM}}$. Importantly, we quantitatively show that there are many Type I adversarial examples, which are classified as successful by $\mathcal{F}_{\mathrm{Curr}}$ but do not to capture true visual identity. The false positive rates is consistently high and up to 99% (See Tab. 3). This demonstrates that many SOTA MI methods report inflated attack accuracy, indicating that actual privacy leakage is significantly lower than previously believed.

**Experimental Setups.** Using our evaluation framework $\mathcal{F}_{\mathrm{MLLM}}$, we reassess 27 SOTA MI attacks across 5 attacks, 4 defenses, 3 private datasets, 4 public datasets, and 9 target models $T$, following the original setups. Detailed settings are in Appx. B.4. We will release the MI-reconstructed image collection publicly upon publication.

**Significant False Positive by $\mathcal{F}_{Curr}$ are Type I adversarial examples.** An ideally successful attack, $x_y^r$ should capture the visual identity of $y$. However, *for a successful attack as according to $\mathcal{F}_{Curr}$, $x_y^r$ only needs to be classified as $y$ by an evaluation model $E$.* As shown in Fig. 1, we observe that within MI-reconstructed images $x_y^r$, there are cases where the visual identity to $y$ is minimal. Nevertheless, $E$ assigns very high probabilities to $y$ for these examples, i.e., high values of $P_E(y|x_y^r)$. We refer to these cases as false positives under the $\mathcal{F}_{Curr}$ framework. To better understand the extent of this false positive rate, we compare the ground truth success rate (established using our $\mathcal{F}_{MLLM}$) to the success rate as measured by $\mathcal{F}_{Curr}$ framework. Particularly, given the MLLM-annotated labels and the prediction via $\mathcal{F}_{Curr}$, we compute the False Positives (FP) rate, False Negatives (FN) rate, True Positives (TP) rate, and True Negatives (TP) rate for each MI setup. The AttAcc via $\mathcal{F}_{Curr}$, $\mathrm{AttAcc}_{\mathcal{F}_{Curr}} = \frac{FP+TP}{FN+TP+FP+TN}$

The results in Tab. 3 consistently show that the FP rates are significant high across MI setups. In other words, there are numerous MI reconstructed images that do not capture visual identity of, yet they are deemed success by $\mathcal{F}_{Curr}$. Such high FP rate contributes to the significant inflation in

Table 3: **Our investigation on MI evaluation framework using our comprehensive dataset of MI attack samples.** We study 27 standard MI setups covering SOTA MI studies (PPA (Struppek et al., 2022), LOMMA (Qiu et al., 2024), KEDMI (Chen et al., 2021), PLGMI (Yuan et al., 2023), IFGMI (Qiu et al., 2024), TL (Ho et al., 2024), TTS (Koh et al., 2024), RoLSS (Koh et al., 2024), LS (Struppek et al., 2024b)), spanning **9** target classifiers $T$, **3** private datasets $\mathcal{D}_{priv}$, and **4** public datasets $\mathcal{D}_{pub}$. Details are in the Appx. B.4. Under $\mathcal{F}_{Curr}$, we find high false positive (FP) rates, indicating that prior work overestimates MI threats. For example, while SOTA attacks (e.g., IFGMI, LOMMA, PLGMI, PPA) claim over 90–100% AttAcc in some setups, actual privacy leakage remains below 80% across all setups, with some attacks falling under 60%.

| MI Attack | $\mathcal{D}_{pub}$ | $\mathcal{D}_{pub}$ | $T$ | $\mathcal{F}_{MLLM}$ AttAcc | $E$ | $\mathcal{F}_{Curr}$ AttAcc | FP rate | FN rate | TP rate | TN rate |
|---|---|---|---|---|---|---|---|---|---|---|
| PPA | FaceScrub | FFHQ | ResNet18 | 28.37% | InceptionNetV3 | 91.39% | 90.09% | 5.32% | 94.58% | 9.91% |
| | | | ResNet101 | 28.68% | | 84.69% | 82.71% | 10.36% | 89.64% | 17.29% |
| | | | ResNet152 | 30.26% | | 86.84% | 85.09% | 9.12% | 90.88% | 14.91% |
| | | | MobileNet-V2 | 47.18% | | 83.37% | 80.39% | 13.30% | 86.70% | 19.61% |
| | | | DenseNet121 | 27.43% | | 72.41% | 70.13% | 21.58% | 78.42% | 29.87% |
| | | | MaxViT | 30.19% | | 79.48% | 77.16% | 15.16% | 84.84% | 22.84% |
| | Stanford Dogs | AFHQ | ResNest101 | 74.58% | InceptionNetV3 | 81.98% | 61.07% | 10.89% | 89.11% | 38.93% |
| IFGMI | FaceScrub | FFHQ | ResNet18 | 34.46% | InceptionNetV3 | 95.85% | 94.60% | 1.78% | 98.22% | 5.40% |
| | | Metfaces | ResNet18 | 1.56% | | 72.50% | 72.21% | 9.09% | 90.91% | 27.79% |
| PLGMI | CelebA | CelebA | VGG16 | 73.73% | FaceNet112 | 98.73% | 99.49% | 1.54% | 98.46% | 0.51% |
| | | FFHQ | VGG16 | 48.47% | | 88.67% | 88.49% | 11.14% | 88.86% | 11.51% |
| LOMMA | CelebA | CelebA | IR152 | 79.80% | FaceNet112 | 90.40% | 86.80% | 8.69% | 91.31% | 13.20% |
| | | | FaceNet64 | 78.73% | | 92.00% | 93.73% | 8.47% | 91.53% | 6.27% |
| | | | VGG16 | 79.93% | | 90.13% | 90.70% | 10.01% | 89.99% | 9.30% |
| | CelebA | FFHQ | IR152 | 44.93% | | 77.73% | 77.85% | 22.40% | 77.60% | 30.27% |
| | | | FaceNet64 | 46.27% | | 72.13% | 69.73% | 25.07% | 74.93% | 22.15% |
| | | | VGG16 | 55.27% | | 63.07% | 61.55% | 35.71% | 64.29% | 38.45% |
| KEDMI | CelebA | CelebA | IR152 | 66.73% | FaceNet112 | 79.27% | 74.55% | 18.38% | 81.62% | 24.45% |
| | | | FaceNet64 | 65.73% | | 80.53% | 78.40% | 18.36% | 81.64% | 21.60% |
| | | | VGG16 | 69.53% | | 73.13% | 69.80% | 25.41% | 74.59% | 30.20% |
| | CelebA | FFHQ | IR152 | 37.67% | | 52.20% | 51.02% | 45.84% | 54.16% | 48.98% |
| | | | FaceNet64 | 36.07% | | 54.60% | 52.24% | 41.22% | 58.78% | 47.76% |
| | | | VGG16 | 38.07% | | 42.47% | 41.33% | 55.69% | 44.31% | 58.67% |

reported AttAcc via $\mathcal{F}_{Curr}$ of latest SOTA MI attack such as PPA, PLGMI, IFGMI, or LOMMA. Notably, in their reported AttAcc using $\mathcal{F}_{Curr}$, these recent attacks report AttAcc values exceeding 90%, or even nearly 100% for certain setups. However, across a wide range of MI setups, the actual success rates never reach 80%. While we focus on high FP rates, FN rates also reveal limitations in the $\mathcal{F}_{Curr}$. Across MI setups, FN rates are consistently lower than FP rates. The FN rates depend on the classification accuracy and generalization capability of $E$. For example, under the PLGMI attack, when $E$ = FaceNet112 is trained on CelebA and evaluated with MI reconstructed images also from CelebA prior ($\mathcal{D}_{pub}$ = CelebA), FN rates are lower. In contrast, if this $E$ is evaluated with MI reconstructed images from FFHQ prior ($\mathcal{D}_{pub}$ = FFHQ), FN rates increase due to distribution shifts.

Furthermore, in certain MI setups, we find that $\mathcal{F}_{Curr}$ *does not align well with* $\mathcal{F}_{MLLM}$ *in evaluating MI attacks.* For example, in the setup of MaxViT as $T$ under the PPA attack, the AttAcc measured by $\mathcal{F}_{Curr}$ is 11.91% lower than the setup for ResNet18 as $T$ under the same attack. However, the MaxViT as $T$ setup shows a 1.82% higher AttAcc measured by $\mathcal{F}_{MLLM}$ than the ResNet18 as $T$ setup. This suggests that, although less effective, the common MI evaluation framework $\mathcal{F}_{Curr}$ could rate the attack as more successful than it actually is. In conclusion, our analysis shows that *the common MI evaluation framework* $\mathcal{F}_{Curr}$ *is suffered from very high FP rate, significantly affecting the reported results of contemporary MI studies based on* $\mathcal{F}_{Curr}$.

Table 4: **Our controlled experiment to show the effect of Type I adversarial attacks in MI and false positive rates.** We compare false positive rates of negative $x_y^r$ (many are affected by Type I adversarial features) and negative $x_y^{natural}$ (free from such features; see main text for construction). Results show $x_y^r$ suffers significantly high false positive rates. Experiments are conducted across all previous MI evaluation models $E$ (e.g., InceptionNetV3 for PPA, FaceNet112 for PLGMI) and an additional architecture (MaxViT for PPA). Further results are provided in the Appx. A.3

| Attack | $\mathcal{D}_{priv}$ | $E$ | | FP rates under $E$ |
|---|---|---|---|---|
| PPA | FaceScrub | InceptionV3 | Neg $x_y^r$ | 90.09% |
| | | | Neg $x_y^{natural}$ | 0.94% |
| | | MaxViT | Neg $x_y^r$ | 73.95% |
| | | | Neg $x_y^{natural}$ | 0.22% |
| PLGMI | CelebA | FaceNet112 | Neg $x_y^r$ | 99.49% |
| | | | Neg $x_y^{natural}$ | 0.00% |

### 4.3 THE EFFECT OF TYPE I ADVERSARIAL FEATURES ON MI IN PRODUCING FALSE POSITIVES

In Sec. 3, we provide an analysis demonstrating that due to Type I adversarial transferability, there are many false positives under the evaluation of $E$. In this section, we provide quantitative results to support our analysis in MI setups.

Due to the similarity nature of Type I adversarial attacks and MI, i.e. maximizing the likelihood with respect to (w.r.t) input under a fixed model, we hypothesize that a significant number of MI reconstructed samples $x_y^r$ carrying Type I adversarial features. As a result, they are mis-classified by $T$ and $E$ into the target label $y$ although these $x_y^r$ do not resemble $x_y$ (Eq. 5). We conduct experiments to validate our hypothesis. We take the two MI setups: ● Setup 1: $T$= ResNet18, $\mathcal{D}_{priv}$= FaceScrub, attack = PPA, $E$= InceptionV3/ MaxVIT ● Setup 2: $T$= VGG16, $\mathcal{D}_{priv}$= CelebA, attack = PLGMI, $E$= FaceNet112. We conduct this experiment on a comprehensive setting with all evaluation models $E$ used in previous MI studies (i.e., InceptionNetV3 for PPA on and FaceNet112 for PLGMI) and an additional architecture of $E$ (i.e., MaxViT for PPA). We first perform MI attacks to obtain MI-generated samples $x_y^r$. We then identified all **MI-generated negative samples**, denoted as Neg $x_y^r$, through MLLM annotation, i.e., $f_{MLLM}(\text{Neg } x_y^r) \neq f_{MLLM}(x_y)$. Let $n$ denote the number of MI-generated negative samples $|\text{Neg } x_y^r| = n$. We create another dataset of **natural negative samples**, Neg $x_y^{natural}$, which is free from Type I adversarial attack, for controlled experiments. We do so by randomly selecting $n$ FFHQ images (no class overlapping with FaceScrub), and intentionally mis-label these FFHQ images with randomly selected FaceScrub identities, obtaining $n$ Neg $x_y^{natural}$ ($f_{MLLM}(\text{Neg } x_y^{natural}) \neq f_{MLLM}(x_y)$). Importantly, unlike Neg $x_y^r$, these Neg $x_y^{natural}$ are randomly selected from FFHQ and therefore they are free from Type I adversarial attack. Neg $x_y^{natural}$ is our controlled dataset.

We pass Neg $x_y^r$ and Neg $x_y^{natural}$ into $E$, and count the number of false positive, i.e., Neg $x_y^r$/ Neg $x_y^{natural}$ being mis-classified into $y$. The false positive rate of Neg $x_y^r$ and Neg $x_y^{natural}$ are compared in Tab. S.5. MI-generated negatives Neg $x_y^r$, with Type I adversarial feature learned during MI, have a high FP rate, whereas natural negatives Neg $x_y^{natural}$, which are free from Type I adversarial feature, have a low FP rate. For example, in Setup 1, FP rates of Neg $x_y^r$ is 90.09% while FP rates of Neg $x_y^{natural}$ is 0.94%. This experiment further demonstrates the **effect of Type I Adversarial features in MI evaluation resulting in a significant number of false positives.**

## 5 CONCLUSION

This work identifies a critical issue in the standard evaluation framework for Model Inversion attacks: the inflation of attack success due to Type I adversarial examples that do not capture true visual identity. To address this, we propose a reliable MLLM-based MI evaluation framework that minimize the impact of Type I adversarial transferability. Our extensive empirical analysis across 27 MI attack setups demonstrates that false positive rates under the standard evaluation framework can reach up to 99%, severely overstating actual privacy leakage. With our proposed evaluation framework, we offer a more accurate and robust way to measure MI attack success, setting a new standard for evaluating privacy risks in machine learning systems. **Limitation and Ethical statement are included in App**

## REPRODUCIBILITY STATEMENT

To ensure the reproducibility of our results, we will make our code and datasets publicly available upon publication. The details of our model architecture, experimental setup, and hyperparameters are provided in the main paper and further elaborated in the appendix. This approach allows other researchers to replicate our experiments and build upon our findings.

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

# Appendix

Contents

# A  ADDITIONAL RESULTS

## A.1  ADDITIONAL RESULTS ON DESIGNING AND IMPLEMENTING $\mathcal{F}_{MLLM}$

### A.1.1  $\mathcal{F}_{MLLM}$ ALIGNS WELL WITH HUMAN EVALUATION

In $\mathcal{F}_{MLLM}$, we employ Gemini to assess the success of MI attacks given a query. In the main paper, we demonstrate that Gemini is effective in recognizing samples from the private dataset. Such experiment is conducted with natural images (i.e., training set of Facescrub). In this Supp, we further demonstrate this with MI reconstructed images.

**Setup.** To establish positive and negative pairs for MI reconstructed images, we leverage human annotation for them. Since human annotation is costly and time-consuming, we sample 30 images per attack setup across 10 setups spanning 5 different MI attack methods. This results in a total of 300 images. Each image is independently evaluated by four human participants. To mitigate the subjectivity of human evaluation, we retain only the images with high inter-annotator agreement, defined as at least 3 out of 4 consistent annotations. The final label for each retained image is the majority vote among the consistent annotations. After filtering, our human-annotated dataset includes 215 images, which we treat as ground truth to assess the reliability of $\mathcal{F}_{\text{MLLM}}$.

**Results.** The results are presented in Tab. S.1. We observe consistently high "Yes" rates for positive pairs and high "No" rates for negative pairs across datasets. This indicates that Gemini is effective at recognizing samples from the private dataset in both natural and MI-reconstructed images. These results further demonstrate that Gemini serves as a reliable evaluator for our MI setups.

Table S.1:  We conduct an experiment to demonstrate Gemini's effectiveness in recognizing samples from the private dataset. This results establish that Gemini can serve as a reliable evaluator in MI attack setups. We collect samples for these data from a comprehensive set of MI setups spanning 5 different MI attacks: PPA (Struppek et al., 2022), IFGMI (Qiu et al., 2024), LOMMA (Nguyen et al., 2023a), KEDMI (Chen et al., 2021), and PLGMI (Yuan et al., 2023), 3 $\mathcal{D}_{pub}$, 2 $\mathcal{D}_{priv}$, and 8 $T$. The details of annotation can be found in Sec. A.1.1.

|  | "Yes" Rate | "No" Rate |
| --- | --- | --- |
| Positive Pair | 95.16% | 4.84% |
| Negative Pair | 22.88% | 77.12% |

### A.1.2  CHATGPT-5 REFUSES TO MI QUERIES

Despite ChatGPT-5 is a powerful closed-source model, it refuses to assess MI queries with high "Refuse" rates. Some examples are provided in Fig. S.1.

### A.1.3  $\mathcal{F}_{MLLM}$ IS ROBUST TO MI EVALUATION ACROSS DATASETS

The results in Tab. S.3 show that $\mathcal{F}_{MLLM}$ is robust to MI evaluation across commonly used dataset in MI research including Facescrub and CelebA

### A.1.4  $\mathcal{F}_{MLLM}$ IS ROBUST TO MI EVALUATION ACROSS PROMPTS

In this section, we provide an analysis of the variance in our proposed framework with respect to: **(1) the choice of reference images**, and **(2) different questions**, as shown in Fig. S.2. For different questions, We run our evaluation framework

Table S.2: We conduct an experiment to demonstrate Gemini's effectiveness in recognizing samples from the private dataset. This results establish that Gemini can serve as a reliable evaluator in MI attack setups.

|  | Dataset | "Yes" rate (%) | "No" rate (%) |
| --- | --- | --- | --- |
| Positive pairs | CelebA | 94.88 | 5.12 |
|  | FaceScrub | 93.84 | 3.16 |
| Negative pairs | CelebA | 8.25 | 91.75 |
|  | FaceScrub | 4.41 | 95.59 |

three times with three different questions: "Does Image A depict the same individual as the images in Set B?", "Does Image A show the same person as those in Set B?", "Is the person in Image A the same as the one(s) shown in Set B?". For different choices of reference images, we run our evaluation

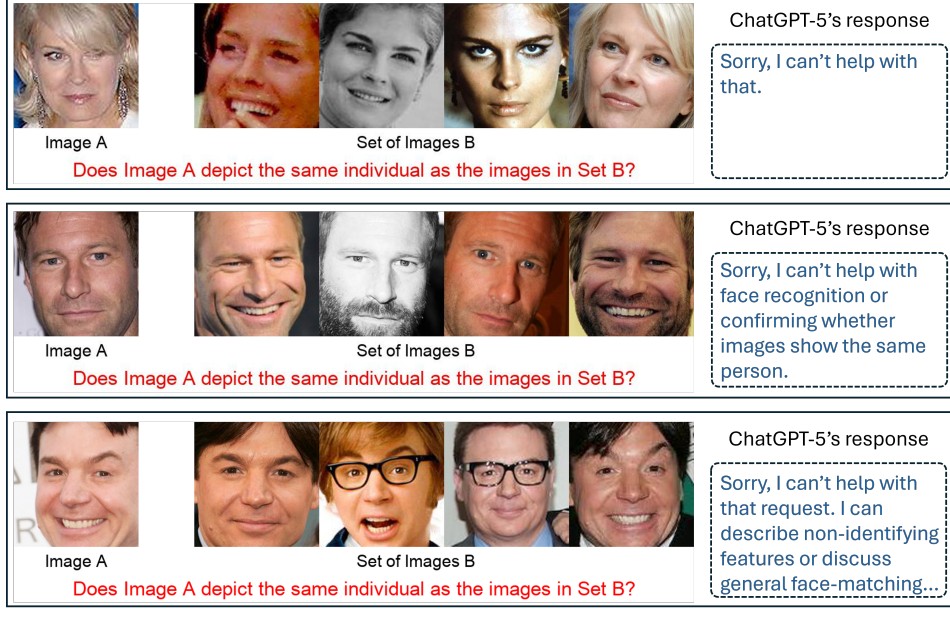

Figure S.1: Examples of ChatGPT-5 refusing to evaluate MI-related queries

framework three times, each with a different random selection of reference images. we show that our $\mathcal{F}_{MLLM}$ is robust to MI evaluation across prompts.

## A.2 EVALUATION RESULTS ON MI DEFENSES

Our main focus in this work is MI attacks, where we highlight that the previously reported success rates using the $\mathcal{F}_{Curr}$ are problematic. In fact, the threat of MI attacks has been overestimated and the amount of leaked information is considerably less than previously assumed. As recent MI defenses also use $\mathcal{F}_{Curr}$ to compute MI success rates, we aim to assess the effectiveness of these defenses using our MLLM-annotated dataset.

Table S.3: $\mathcal{F}_{MLLM}$ is robust to MI evaluation across prompts

|  | Dataset | "Yes" rate (%) | "No" rate (%) |
|---|---|---|---|
| Positive pairs | CelebA | 94.88 | 5.12 |
| | FaceScrub | 93.84 | 3.16 |
| Negative pairs | CelebA | 8.25 | 91.75 |
| | FaceScrub | 4.41 | 95.59 |

In this section, we focus on high-resolution setups with PPA (Struppek et al., 2022). Specifically, we include the latest SOTA MI defenses, such as RoLSS (Koh et al., 2024), TL (Ho et al., 2024), LS (Struppek et al., 2024a), and TTS (Koh et al., 2024). The MI setups strictly follow the configurations in these MI defense studies. The results can be found in Tab. S.4.

In general, similar to our observations on MI attacks in the main manuscript, $\mathcal{F}_{Curr}$ may inaccurately assess the effectiveness of SOTA MI defenses. For example, we observe a mismatch between AttAcc comparisons via $\mathcal{F}_{Curr}$ and AttAcc measured by $\mathcal{F}_{MLLM}$. For example, AttAcc via $\mathcal{F}_{Curr}$ suggests that TL (Ho et al., 2024) outperforms RoLSS and TTS (Koh et al., 2024). However, AttAcc via $\mathcal{F}_{MLLM}$ indicates that RoLSS and TTS are more effective defenses. In what follows, we further discuss these results.

These MI defenses result in a reduction in FP rates due to the degradation of the transferability of adversarial characteristics from $T$ to $E$. Specifically, under TL defense (Ho et al., 2024), only the later layers of $T$ are fine-tuned on $\mathcal{D}_{priv}$, while earlier layers are frozen from the pre-trained backbone. Hence, later layers of $T$ capture $\mathcal{D}_{priv}$ features, while earlier layers of $T$ capture $\mathcal{D}_{pretrain}$ features. In contrast, $E$ captures $\mathcal{D}_{priv}$ features across all layers since all layers of $E$ are fine-tuned on $\mathcal{D}_{priv}$. This mismatch in feature representations between $T$ and $E$ under TL is likely to reduce adversarial

Table S.4: **Our investigation on the effectiveness of MI defenses using our MLLM-annotated dataset of MI attack samples.** We present the results of the latest MI defenses including RoLSS (Koh et al., 2024), TL (Ho et al., 2024), LS (Struppek et al., 2024a), and TTS (Koh et al., 2024). We observe a mismatch between AttAcc comparisons via $\mathcal{F}_{Curr}$ and actual AttAcc measured by $\mathcal{F}_{MLLM}$. Overall, consistent with our findings on MI attacks, this suggests that $\mathcal{F}_{Curr}$ may have issues in evaluating MI defenses.

| $T$ | Model Acc | $\mathcal{F}_{MLLM}$ AttAcc | $\mathcal{F}_{Curr}$ | | | | | |
|---|---|---|---|---|---|---|---|---|
| | | | $E$ | AttAcc | FP rate | FN rate | TP rate | TN rate |
| ResNet101 | 94.86% | 28.68% | | 84.69% | 82.71% | 10.36% | 89.64% | 17.29% |
| ResNet101-RoLSS | 92.98% | 19.46% | | 43.47% | 40.70% | 45.09% | 54.91% | 59.30% |
| ResNet101-TL | 92.51% | 25.09% | InceptionNetV3 | 34.17% | 31.27% | 57.14% | 42.86% | 68.73% |
| ResNet101-TTS | 94.16% | 18.44% | | 42.52% | 39.39% | 43.61% | 56.39% | 60.61% |
| ResNet101-LS | 92.21% | 10.54% | | 16.56% | 14.90% | 69.35% | 30.65% | 85.10% |

transferability (Ilyas et al., 2019; Qin et al., 2022; Ma et al., 2024), thereby reducing FP rates. Under LS defense (Struppek et al., 2024a), negative label smoothing (LS) is employed to improve MI robustness. LS slightly reduces label dominance and weakens gradient alignment between surrogate and target models (Zhang et al., 2024). Negative LS amplifies this effect, further degrading gradient similarity. Therefore, training $T$ with negative LS diminishes gradient alignment with $E$ (trained on standard labels), reducing adversarial transferability (Zhang et al., 2024; Demontis et al., 2019). Under RoLSS and TTS defenses (Koh et al., 2024), removing certain skip connections improves resilience to MI attacks. Skip connections are known to improve adversarial transferability (Wu et al., 2020). By modifying $T$ to remove some skip connections, adversarial examples generated by $T$ transfer less effectively to $E$.

Regarding FN rates, although this is not the main focus of our study, we observe that FN rates tend to increase under MI defenses compared to MI attacks. FN rates depend on the classification accuracy and generalization capability of $E$. SOTA MI defenses introduce various strategies (e.g., fixing earlier layers trained on public data (Ho et al., 2024), perturbing labels (Struppek et al., 2024a), and removing skip connections (Koh et al., 2024)) to encode less information in the predictions of $T$. These approaches may encourage $T$ to learn more generalized features. As a result, reconstructed images based on these generalized features of $T$ may differ more from the seen training data. However, in the prevalent MI setups, $E$ in $\mathcal{F}_{Curr}$ is often trained with standard training procedures and architectures. This could limit its generalization capacity, making it less capable of accurately classifying these reconstructed images via the target models $T$ under MI defenses.

### A.3 ADDITIONAL RESULTS ON THE EFFECT OF TYPE I ADVERSARIAL ATTACKS IN MI ON FALSE POSITIVE RATES

In Sec. 4.2 in the main manuscript, we provide an analysis to demonstrate the effect of Type I adversarial attacks in MI on false positive rates. In this Supp, we provide results on additional setups. The results are presented in Tab. S.5. These additional results are consistent with our observation in the main manuscript demonstrating the effect of Type I Adversarial features in MI evaluation resulting in a significant number of false positives.

## B DETAILED EXPERIMENTAL REPRODUCIBILITY

### B.1 DETAILED $\mathcal{F}_{MLLM}$ SETUP

Our implementation of $\mathcal{F}_{MLLM}$ is illustrated in Fig. S.2. To evaluate whether a reconstructed image is a successful or unsuccessful attack, we employ Gemini 2.0 Flash API (see the main manuscript for our justification for choosing Gemini) for the evaluation.

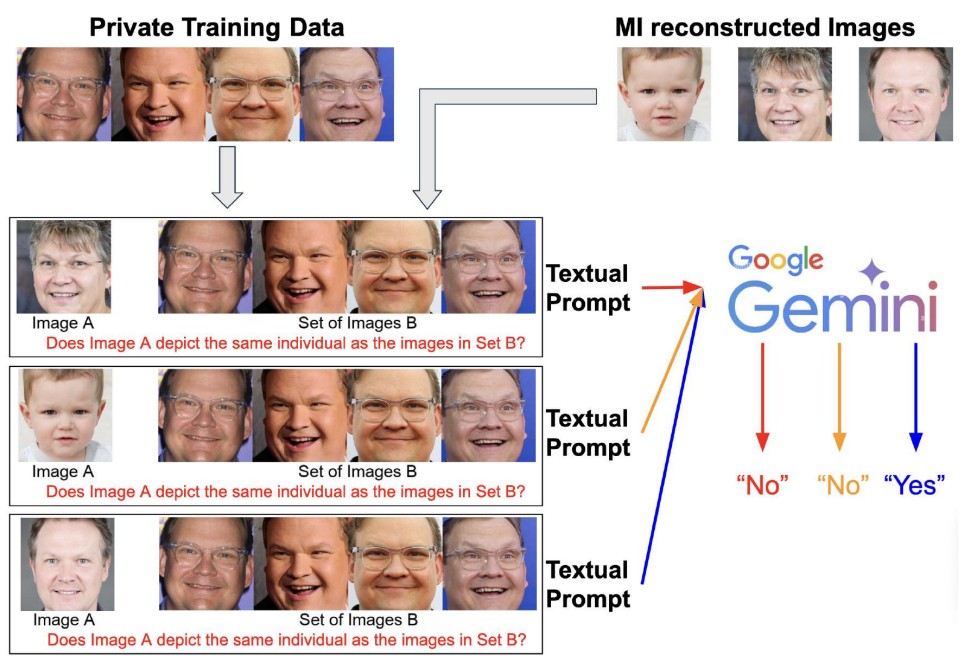

Figure S.2: **Our detailed implementation of MLLM-based MI Evaluation Framework $\mathcal{F}_{MLLM}$.** For each reconstructed image, we pair with a set of private training data to construct an evaluation query image. Then, each evaluation query image is passed to Gemini with a textual prompt. The detailed of textual prompt can be found in Sec. B.2. The final attack accuracy is computed based on Gemini's responses.

Given a reconstructed image (Image A), we construct an evaluation query image by pairing it with a set of private training images (Set B) that includes the target identity. We then formulate a natural language textual prompt along with the evaluation query image and pass it to Gemini. The textual prompts are shown in the table below and are fixed across evaluation queries for a fair comparison.

For each reconstructed image, the model outputs a categorical response ("Yes" or "No"). A "Yes" answer is interpreted as a successful attack. By evaluating a large number of such queries and computing the proportion of correct identifications, $\mathcal{F}_{MLLM}$ provide an automated and faithful evaluation of MI.

### B.2 THE DETAILED PROMPT IN $\mathcal{F}_{MLLM}$

The detailed textual prompts in our MI evaluation framework can be found in Tab. S.6.

### B.3 ERROR BAR OF EVALUATION RESULTS WITH $\mathcal{F}_{MLLM}$

As mentioned in the main manuscript, we provide an error bar of evaluation results with $\mathcal{F}_{MLLM}$ to further demonstrate the robustness of our proposed MI evaluation framework. The results can be found in Tab. S.7

### B.4 DETAILED MI SETUP

To ensure the reproducibility, we strictly follow previous studies (Zhang et al., 2020; Chen et al., 2021; Nguyen et al., 2023a; Struppek et al., 2022; Qiu et al., 2024; Koh et al., 2024; Ho et al., 2024) for MI setups.

**MI attacks.** Our study focuses on SOTA GAN-based MI attack that achieve strong performance in computer vision domain. These attacks optimize the GAN latent space rather than directly optimize the image space.

Table S.5: **Our controlled experiment to show the effect of Type I adversarial attacks in MI on false positive rates.** We provide results on this experiment on additional setups in Appx. B.4

| Attack | $E$ | $\mathcal{D}_{priv}$ | $\mathcal{D}_{pub}$ | $T$ | | FP rates under $E$ |
|---|---|---|---|---|---|---|
| PPA | InceptionV3 | Facescrub | FFHQ | Resnet101 | Neg $x_y^r$ | 82.71% |
| | | | | | Neg $x_y^{natural}$ | 0.94% |
| | | | | Resnet152 | Neg $x_y^r$ | 85.09% |
| | | | | | Neg $x_y^{natural}$ | 0.94% |
| | | | | MaxViT | Neg $x_y^r$ | 79.48% |
| | | | | | Neg $x_y^{natural}$ | 0.94% |
| | | | | DenseNet121 | Neg $x_y^r$ | 72.41% |
| | | | | | Neg $x_y^{natural}$ | 0.94% |
| IFGMI | | | MetFaces | Resnet18 | Neg $x_y^r$ | 72.71% |
| | | | | | Neg $x_y^{natural}$ | 0.94% |
| PLGMI | | | FFHQ | VGG16 | Neg $x_y^r$ | 88.49% |
| | | | | | Neg $x_y^{natural}$ | 0.00% |
| LOMMA | FaceNet112 | CelebA | CelebA | FaceNet64 | Neg $x_y^r$ | 93.73% |
| | | | | | Neg $x_y^{natural}$ | 0.00% |
| | | | | IR152 | Neg $x_y^r$ | 86.80% |
| | | | | | Neg $x_y^{natural}$ | 0.00% |
| | | | | VGG16 | Neg $x_y^r$ | 90.70% |
| | | | | | Neg $x_y^{natural}$ | 0.00% |
| | | | FFHQ | FaceNet64 | Neg $x_y^r$ | 69.73% |
| | | | | | Neg $x_y^{natural}$ | 0.00% |
| | | | | IR152 | Neg $x_y^r$ | 77.85% |
| | | | | | Neg $x_y^{natural}$ | 0.00% |
| | | | | VGG16 | Neg $x_y^r$ | 61.55% |
| | | | | | Neg $x_y^{natural}$ | 0.00% |
| KEDMI | | | CelebA | FaceNet64 | Neg $x_y^r$ | 78.40% |
| | | | | | Neg $x_y^{natural}$ | 0.00% |
| | | | | IR152 | Neg $x_y^r$ | 74.55% |
| | | | | | Neg $x_y^{natural}$ | 0.00% |
| | | | | VGG16 | Neg $x_y^r$ | 69.80% |
| | | | | | Neg $x_y^{natural}$ | 0.00% |
| | | | FFHQ | FaceNet64 | Neg $x_y^r$ | 52.24% |
| | | | | | Neg $x_y^{natural}$ | 0.00% |
| | | | | IR152 | Neg $x_y^r$ | 51.02% |
| | | | | | Neg $x_y^{natural}$ | 0.00% |
| | | | | VGG16 | Neg $x_y^r$ | 41.33% |
| | | | | | Neg $x_y^{natural}$ | 0.00% |

*KEDMI* (Chen et al., 2021) Introduces an MI-specific GAN that incorporates knowledge from the target classifier. The discriminator performs dual tasks: distinguishing real and fake samples and predicting class-wise labels.

*LOMMA* (Nguyen et al., 2023a) Improves MI attacks using a novel logit loss and model augmentation to mitigate overfitting.

*PLGMI* (Yuan et al., 2023) Leverages conditional GANs to isolate class-specific search spaces and uses Max-Margin Loss to address vanishing gradients in MI optimization.

*PPA* (Struppek et al., 2022) Utilizes powerful StyleGAN for high-resolution image MI attacks, emphasizing a modular design adaptable to different architectures and datasets.

*IFGMI* (Qiu et al., 2024) Proposes Intermediate Features Generative Model Inversion, extending optimization from latent codes to intermediate features, enhancing the attack's expressive capability.

Table S.6: Detailed textual prompts in our MI evaluation framework $\mathcal{F}_{MLLM}$

| Domain | $\mathcal{D}_{priv}$ | Textual Prompt |
|---|---|---|
| Face Recognition | Facescrub 
 ———— 
 CelebA | You are an expert in face recognition. Taking into account the face aging, lighting, different hair styles, wearing and not wearing of eye glasses or other accessory, do the task in the image. Only answer yes or no |
| Dog Recognition | Stanford Dogs | You are an expert in dog breed recognition. Taking into account the lighting and viewpoint, do the task in the image. Only answer yes or no |

**MI defense.**  Our study focuses on SOTA MI defenses. Differ from MI attacks, MI defenses aim to minimize the disclosure of training samples during the MI optimization process.

*TL (Ho et al., 2024)* Leverages Transfer Learning to limit sensitive information encoding in earlier layers, degrading MI attack performance.

*LS (Struppek et al., 2024a)* Introduces label smoothing with negative factors to impede class-related information extraction.

*RoLSS (Koh et al., 2024)* Demonstrates that removing skip connections in the last stage significantly reduces MI attack accuracy, offering a better MI robustness trade-off.

*TTS (Koh et al., 2024)* Buliding on top of RoLSS. Particularly, in the first stage, the model $T$ with full skip-connections architecture is trained on private dataset. Then in the stage 2, the skip connection removed architecture, i.e. RoLSS, is fine-tuned on private dataset. The pre-trained parameters in Stage 1 serves as initialization for the stage 2, thereby improve the convergence of model in stage 2.

**Private training data $\mathcal{D}_{priv}$.**  Following previous works (Zhang et al., 2020; Chen et al., 2021; Nguyen et al., 2023a; Struppek et al., 2022; Qiu et al., 2024; Koh et al., 2024; Ho et al., 2024), we focus on reconstruction of images and use the face recognition as a running example including FaceScrub (Ng & Winkler, 2014) and CelebA (Liu et al., 2015).

*FaceScrub* (Ng & Winkler, 2014): FaceScrub provides cropped facial images for 530 identities. The dataset publicly a total of 37,878 images. After train/test splitting, this resulted in 34,090 training samples and 3,788 test samples.

*CelebA* (Liu et al., 2015): CelebA is a dataset of celebrity facial images available for non-commercial research. Following previous works (Zhang et al., 2020; Chen et al., 2021; Nguyen et al., 2023a; Struppek et al., 2022; Qiu et al., 2024; Koh et al., 2024; Ho et al., 2024), we select the top 1,000 identities with the most samples from 10,177 available identities, resulting in 27,034 training samples and 3,004 test samples.

**Public data for GAN $\mathcal{D}_{pub}$.**  Following the data preparation in previous works (Zhang et al., 2020; Chen et al., 2021; Nguyen et al., 2023a; Struppek et al., 2022; Qiu et al., 2024; Koh et al., 2024; Ho et al., 2024), we use $\mathcal{D}_{pub}$ ensuring that the dataset $\mathcal{D}_{priv}$ and $\mathcal{D}_{pub}$ with no class intersection. $\mathcal{D}_{priv}$ is used to train the target classifier $T$, while $\mathcal{D}_{pub}$ is used to train GAN to extract general features only.

*CelebA* (Liu et al., 2015): Following previous works (Zhang et al., 2020; Chen et al., 2021; Nguyen et al., 2023a; Ho et al., 2024), we select 30,000 images from identities distinct from the 1,000 identities in $\mathcal{D}_{priv}$.

*FFHQ* (Karras et al., 2019): This dataset contains 70,000 high-quality human face images sourced from Flickr, offering significant diversity in age, ethnicity, and backgrounds.

*MetFaces (Karras et al., 2020).* This dataset includes 1,336 high-quality artistic renderings of human faces, covering various art styles. The images exhibit significant diversity and uniqueness.

**Target Classifier $T$.**  Following previous works (Zhang et al., 2020; Chen et al., 2021; Nguyen et al., 2023a; Struppek et al., 2022; Qiu et al., 2024; Koh et al., 2024; Ho et al., 2024), we include

Table S.7: **Our investigation on MI evaluation framework using our comprehensive dataset of MI attack samples.** We run the evaluations with our $\mathcal{F}_{MLLM}$ threes times and report mean $\pm$ std.

| MI Attack | $\mathcal{D}_{pub}$ | $\mathcal{D}_{pub}$ | $T$ | $\mathcal{F}_{MLLM}$ AttAcc | $E$ | $\mathcal{F}_{Curr}$ AttAcc | FP rate | FN rate | TP rate | TN rate |
|---|---|---|---|---|---|---|---|---|---|---|
| PPA | FaceScrub | FFHQ | ResNet18 | 28.22±0.30% | | 91.39% 90.03±0.09% | 4.82±0.51% | 94.82±0.32% | 9.97±0.09% | |
| | | | ResNet101 | 28.48±0.36% | | 84.69% 82.79±0.09% | 10.52±0.16% | 89.48±0.16% | 17.21±0.09% | |
| | | | ResNet152 | 30.20±0.09% | InceptionNetV3 | 86.84% 85.13±0.14% | 9.21±0.34% | 90.79±0.34% | 14.87±0.14% | |
| | | | DenseNet121 | 27.44±0.27% | | 72.41% 70.11±0.05% | 21.52±0.07% | 78.48±0.07% | 29.89±0.05% | |
| | | | MaxViT | 30.30±0.13% | | 79.48% 77.32±0.16% | 15.54±0.33% | 84.46±0.34% | 22.68±0.14% | |
| IFGMI | FaceScrub | FFHQ | ResNet18 | 34.14±0.29% | InceptionNetV3 | 95.85% 94.61±0.03% | 1.75±0.07% | 98.25±0.07% | 5.39±0.03% | |
| | | Metfaces | ResNet18 | 1.57±0.07% | | 72.50% 72.24±0.05% | 11.39±3.74% | 88.61±3.74% | 27.76±0.05% | |
| PLGMI | CelebA | CelebA | VGG16 | 73.51±0.75% | FaceNet112 | 98.73% 99.33±0.14% | 1.48±0.05% | 98.52±0.05% | 0.67±0.14% | |
| | | FFHQ | VGG16 | 48.59±0.57% | | 88.67% 88.51±0.07% | 11.16±0.06% | 88.84±0.06% | 11.47±0.07% | |
| LOMMA | CelebA | CelebA | IR152 | 79.02±0.30% | | 92.00% 92.47±1.22% | 8.13±0.33% | 91.87±0.33% | 7.53±1.22% | |
| | | | FaceNet64 | 79.76±0.27% | | 90.40% 87.71±0.80% | 8.92±0.20% | 91.08±0.20% | 12.29±0.08% | |
| | | | VGG16 | 80.73±0.77% | FaceNet112 | 90.13% 90.29±0.95% | 9.91±0.22% | 90.09±0.21% | 9.71±0.95% | |
| | | FFHQ | IR152 | 45.60±0.58% | | 77.73% 77.37±0.45% | 21.84±0.52% | 78.16±0.52% | 22.63±0.45% | |
| | | | FaceNet64 | 45.49±0.74% | | 72.13% 69.92±0.18% | 25.21±0.17% | 74.79±0.17% | 30.08±0.18% | |
| | | | VGG16 | 56.09±1.20% | | 63.07% 61.33±0.31% | 35.58±0.18% | 64.42±0.18% | 38.67±0.31% | |
| KEDMI | CelebA | CelebA | IR152 | 67.24±0.83% | | 79.27% 74.97±0.37% | 18.64±0.25% | 81.36±0.25% | 24.70±0.23% | |
| | | | FaceNet64 | 66.15±0.73% | | 80.53% 77.27±1.04% | 17.81±0.49% | 82.19±0.49% | 30.15±1.56% | |
| | | | VGG16 | 69.38±1.04% | FaceNet112 | 73.13% 69.85±1.56% | 25.44±0.62% | 74.56±0.62% | 30.15±1.56% | |
| | | FFHQ | IR152 | 36.96±0.62% | | 52.20% 50.24±0.75% | 44.42±1.34% | 55.58±1.34% | 49.76±0.75% | |
| | | | FaceNet64 | 35.96±0.14% | | 54.60% 52.08±0.62% | 40.91±1.13% | 59.09±1.13% | 47.92±0.62% | |
| | | | VGG16 | 38.85±0.80% | | 42.47% 41.24±0.36% | 55.50±0.58% | 44.40±0.58% | 58.76±0.36% | |

a wide ranges of architectures as $T$ in our study including ResNet18/101/152 (He et al., 2016), DenseNet121(Huang et al., 2017), MaxViT (Tu et al., 2022), FaceNet (Chen et al., 2021), and VGG16 (He et al., 2016). To ensure the reproducibility, we utilize the checkpoints of these target classifier in the previous works.

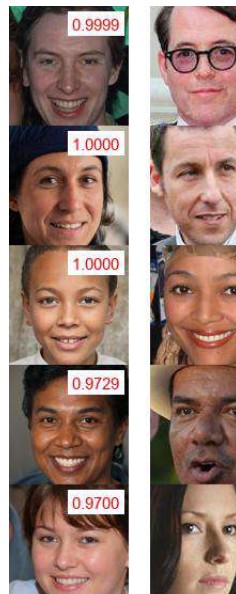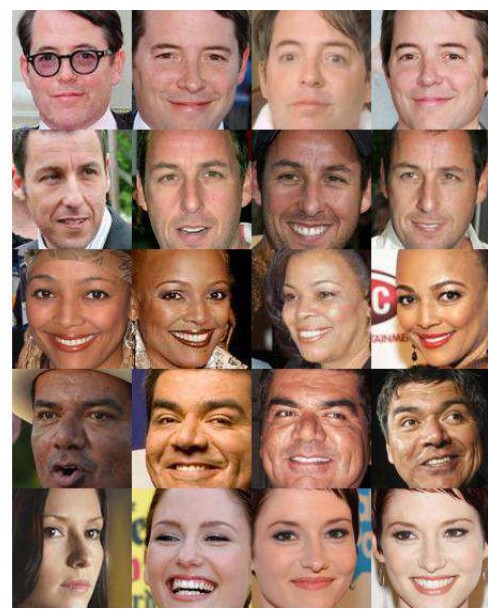

Figure S.3: Additional visualization of false positives. These MI false positives do not capture visual identity features of the target individual in the private training data, but they are still deemed successful attacks according to $\mathcal{F}_{Curr}$ with a high confidence (indicated in red text). Here, $T$=MaxViT (Tu et al., 2022), $\mathcal{D}_{priv}$=FaceScrub (Ng & Winkler, 2014), $\mathcal{D}_{pub}$=FFHQ (Karras et al., 2019), $E$=InceptioNetV3 under PPA attack (Struppek et al., 2022).

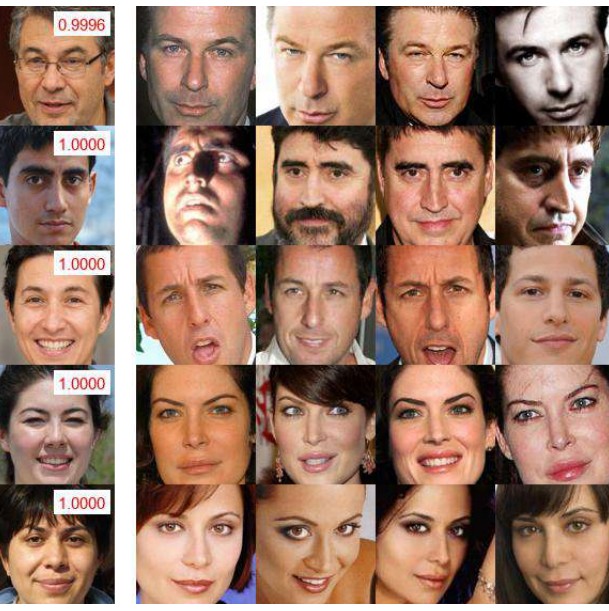

Figure S.4: Additional visualization of false positives. These MI false positives do not capture visual identity features of the target individual in the private training data, but they are still deemed successful attacks according to $\mathcal{F}_{Curr}$ with a high confidence (indicated in red text). Here, $T$=DenseNet121 (Huang et al., 2017), $\mathcal{D}_{priv}$=FaceScrub (Ng & Winkler, 2014), $\mathcal{D}_{pub}$=FFHQ (Karras et al., 2019), $E$=InceptioNetV3 under PPA attack (Struppek et al., 2022).

### B.5 COMPUTING RESOURCES

We conducted all experiments on NVIDIA RTX A5000 GPUs running Ubuntu 20.04.2 LTS, with AMD Ryzen Threadripper PRO 5975WX 32-Core processors. The environment setup includes CUDA 12.2, Python 3.8.18, and PyTorch 1.12.0 with Torchvision 0.14.1. For high-resolution tasks, (Struppek et al., 2022; Qiu et al., 2024), we use model architectures and pre-trained ImageNet backbone weights

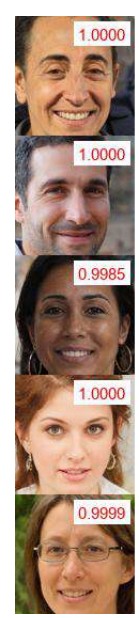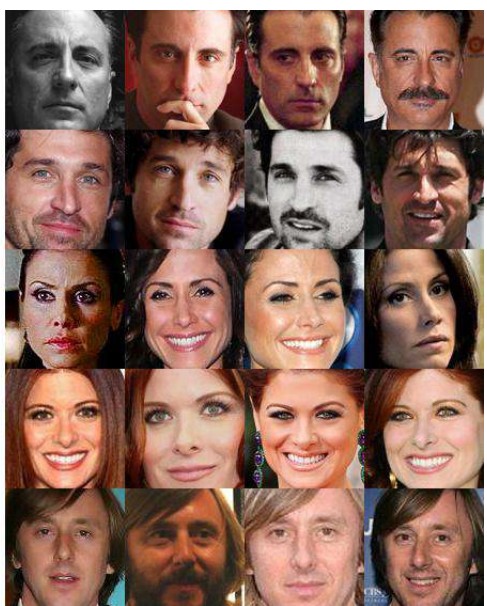

Figure S.5: Additional visualization of false positives. These MI false positives do not capture visual identity features of the target individual in the private training data, but they are still deemed successful attacks according to $\mathcal{F}_{Curr}$ with a high confidence (indicated in red text). Here, $T$=ResNet101 (He et al., 2016), $\mathcal{D}_{priv}$=FaceScrub (Ng & Winkler, 2014), $\mathcal{D}_{pub}$=FFHQ (Karras et al., 2019), $E$=InceptioNetV3 under PPA attack (Struppek et al., 2022).

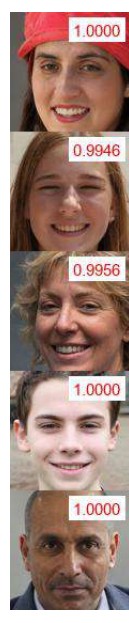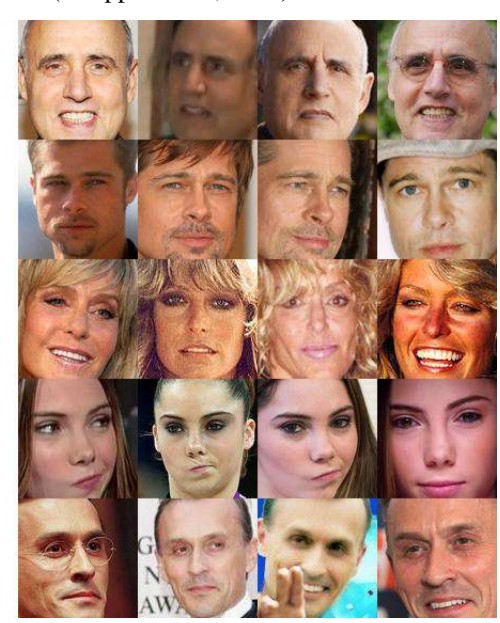

Figure S.6: Additional visualization of false positives. These MI false positives do not capture visual identity features of the target individual in the private training data, but they are still deemed successful attacks according to $\mathcal{F}_{Curr}$ with a high confidence (indicated in red text). Here, $T$=ResNet152 (He et al., 2016), $\mathcal{D}_{priv}$=FaceScrub (Ng & Winkler, 2014), $\mathcal{D}_{pub}$=FFHQ (Karras et al., 2019), $E$=InceptioNetV3 under PPA attack (Struppek et al., 2022).

from Torchvision. For the low-resolution setup, following (Chen et al., 2021; Nguyen et al., 2023a; Ho et al., 2024), we employed VGG architecture with pre-trained ImageNet weights from Torchvision, while we utilize IR152 and FaceNet architectures with pre-trained backbones from face.evoLVe[1].

---

[1]https://github.com/ZhaoJ9014/face.evoLVe

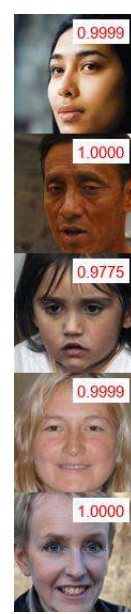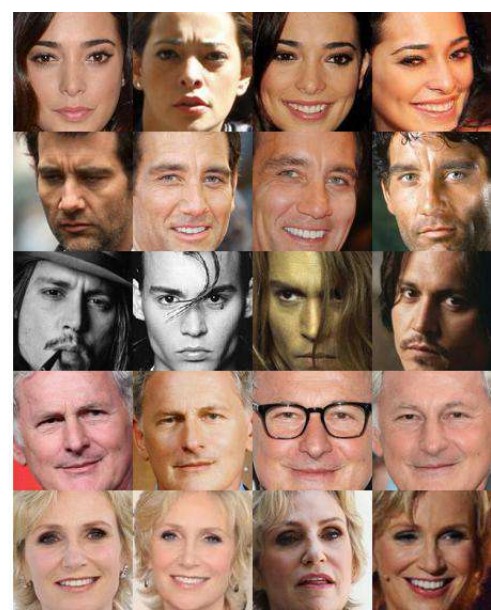

Figure S.7: Additional visualization of false positives. These MI false positives do not capture visual identity features of the target individual in the private training data, but they are still deemed successful attacks according to $\mathcal{F}_{Curr}$ with a high confidence (indicated in red text). Here, $T$=ResNet18 (He et al., 2016), $\mathcal{D}_{priv}$=FaceScrub (Ng & Winkler, 2014), $\mathcal{D}_{pub}$=FFHQ (Karras et al., 2019), $E$=InceptioNetV3 under IFGMI attack (Qiu et al., 2024).

We employ the Gemini 2.0 Flash API in $\mathcal{F}_{\text{MLLM}}$ and emphasize that our implementation is both reliable and cost efficient. Particularly, in our implementation, each evaluation query costs \$0.0002886 (see the official Gemini API documentation[2] for cost estimation). This cost is reasonable for large-scale evaluations. For example, in our study involving larger-scale 26 experimental setups and a total of 71,880 MI-reconstructed images, the overall cost is around \$20.75, making our evaluation framework scalable and accessible for future research.

## C ADDITIONAL VISUALIZATION OF FALSE POSITIVES

In the main paper, we provide some visualizations of MI false positives. In this Supp., we provide more extensive visualizations of MI false positives in Fig. S.3, S.4, S.5, S.6, S.7.

These false positive MI do not capture the visual identity features of the target individual in private training data, but are still considered successful attacks according to $\mathcal{F}_{Curr}$ with high confidence.

## D LIMITATION

While this study provides valuable insights into the limitations of the MI evaluation framework and propose a more reliable automated MI evaluation framework for future MI study, it is important to acknowledge certain limitations. One such limitation is the focus on specific architectures and datasets. While we strictly follow previous works (Zhang et al., 2020; Chen et al., 2021; Nguyen et al., 2023a; Struppek et al., 2022; Qiu et al., 2024; Koh et al., 2024; Ho et al., 2024) to includes 26 MI setups, these setups may not include the latest architectures or dataset that are not considered in prevalent MI setups. Future research could expand upon our findings by exploring a wider range of model architectures and datasets. This would further shed the light of MI evaluation and contribute to the development of better MI evaluation frameworks.

---

[2]https://ai.google.dev/gemini-api/docs

# E    ETHICAL STATEMENT

This study examines the limitations of widely used evaluation frameworks for Model Inversion (MI) attacks, which hold critical implications for privacy and data security. Our analysis reveals an overestimation of MI attack success rates, underscoring the need for accurate and reliable evaluation metrics to avoid inflated perceptions of privacy risks. To support the research community, we propose a more reliable and cost-efficient MI evaluation framework based on MLLM. Furthermore, we release the code and a large-scale collection of MI reconstructed images upon publication, advocating for their ethical use to advance privacy protection.

# F    LLM USAGE

We used a large language model to help polish the grammar, wording, and other minor text issues in this manuscript. The authors are fully responsible for the ideas, analysis and conclusions in this submission.

