# OpenReview forum: "Revisiting Model Inversion Evaluation: From Misleading Standards to Reliable Privacy Assessment"
_ICLR.cc/2026/Conference — ICLR 2026 Conference Withdrawn Submission_

### Official Review · Reviewer_4iWz · 2025-10-15

**Soundness:** 2
**Presentation:** 2
**Contribution:** 2
**Rating:** 4
**Confidence:** 4

**Summary:**

The paper revisits the evaluation metric of model inversion attacks (MIAs) and points out the weakness of *Attack Accuracy* using another classification model as the evaluation model. This paper connects the Type-I error of adversarial attack and the False Positive in model inversion attacks. Then, evaluation by MLLM is proposed.

**Strengths:**

+ The perspective of evaluation bias is novel.
+ The connection between MI and adversarial type-I attack sounds interesting.

**Weaknesses:**

1. Prompt has a significant impact on the accuracy of MLLM evaluation. However, this paper only provides the results with the same prompt. The influence of prompt has not been researched.
2. Existing MIA methods do not deliberately conduct adversarial attacks, but the article lacks specific explanations. There is a lack of theoretical analysis of existing attack methods to improve Type-I errors.
3.  Lack of comparison of the type-I error between $F_{curr}$ and $F_{MLLM}$ .
4. While  $F_{MLLM}$ arguably provides a more faithful perceptual evaluation, it is unclear how its judgment correlates with real-world privacy leakage. MLLMs may focus on semantic coherence rather than sensitive visual features.
5. Will this approach raise type-II error?
6. How do various attack and defense algorithms compare under the new evaluation? The comparison is not shown clearly in the paper.

**Questions:**

See "weakness"

---

### Official Review · Reviewer_2isW · 2025-10-26

**Soundness:** 3
**Presentation:** 3
**Contribution:** 3
**Rating:** 6
**Confidence:** 4

**Summary:**

The paper argues that the standard evaluation protocol for model inversion (MI) attacks, measuring success with the same (or closely related) classifier that was attacked, systematically overestimates privacy leakage because MI often produces Type-I adversarial images that trigger the target classifier without actually capturing the true visual identity. The authors (i) diagnose this failure mode; (ii) propose an alternative evaluator, FMLLM, which uses a multimodal LLM (MLLM) to decide whether a reconstructed image matches the ground-truth identity using interleaved image–text prompts; and (iii) re-evaluate a large suite of recent MI attacks. Across 27 MI setups (9 target classifiers × multiple datasets/attacks), they report very high false-positive rates under FCurr (up to 99%), and show that headline attack accuracies above 90-100% in prior work drop below ~80% (sometimes <60%) when re-evaluated with FMLLM. They further provide controlled experiments contrasting "MI negatives" vs. "natural negatives", showing that the inflated success largely comes from adversarial transfer to the evaluation classifier, not genuine identity recovery. Finally, they propose criteria for selecting a reliable MLLM evaluator and recommend "Gemini 2.0" based on a small validation showing high "Yes" on positive pairs, high "No" on negatives, and low refusal rate.

**Strengths:**

**Sharp diagnosis of evaluator coupling.** The manuscript clearly argues that evaluating MI reconstructions with the same (or closely related) classifier can count adversarially transferrable images as "success", not genuine identity recovery. The transferability phenomenon is well-documented across vision models, so flagging it as a core confounder is compelling.

**Broad, controlled re-evaluation.** Re-running diverse MI attacks across multiple targets and reporting error decompositions (not just a single accuracy) is the right methodology and should recalibrate how the field interprets prior headline numbers. A recent survey underscores how heterogeneous MI setups are, making this breadth especially valuable.

**Practical, scalable judging.** Using a capable M/VLM as an automatic judge is inexpensive and aligns with emerging practice where model-based evaluators can approximate human preferences in visual tasks (e.g., text-to-3D evaluation) while avoiding the logistics of large human studies.

**Likely field impact.** By decoupling the attack target from the evaluator, the work promotes a more defensible standard for reporting privacy leakage, consistent with lessons from adversarial robustness that who evaluates you matters.

**Weaknesses:**

**Single-model judge risk, closed, drifting, and biased.** The manuscript relies on a single proprietary MLLM as the evaluator [1]. Independent studies document position and order biases in LLM-as-a-judge settings, along with verdict instability under mirrored orderings and small prompt changes [2][3]. API models can also drift over time, which undermines reproducibility unless versions and prompts are frozen and multiple judges are reported [9]. Recommended mitigation: report agreement across two or more judges, randomize candidate order, and pre-register prompts.

**Limited human grounding of the evaluator.** The paper's claims hinge on an automatic judge; although it builds a human-annotated MI dataset, the evidence that the judge tracks human decisions at scale is limited [1]. Meta-evaluations show LLM judges can exhibit systematic judgment biases, which argues for a larger human study with inter-rater agreement and targeted error analysis before treating an MLLM as a gold-standard referee for identity matching [4].

**Evaluator reliability varies by domain.** Vision-language models perform unevenly on expert image interpretation. Peer-reviewed studies report strong results on text-only exam items but marked drops on image-based questions across radiology boards and related evaluations, with additional reports of misinterpretation and refusal behavior [6][7][8][5]. Treating a general-purpose MLLM as an identity verifier should therefore include domain-specific checks and refusal-rate reporting.

**Scope mostly faces, uncertain generalization.** Model inversion spans multiple modalities and threat models beyond faces. Recent surveys detail image, text, graph, and federated learning variants, which means conclusions drawn from face-recognition targets alone may not transfer without additional evidence [10][11]. A non-face case study would strengthen external validity.

References:

[1] Ho S-T, Jun Hao K, Nguyen N-B, Binder A, Cheung N-M. Uncovering the Limitations of Model Inversion Evaluation: Benchmarks and Connection to Type-I Adversarial Attacks. arXiv:2505.03519, 2025.

[2] Lu Y et al. Judging the Judges: A Systematic Study of Position Bias in LLM-as-a-Judge. arXiv:2406.07791, 2024.

[3] Chen X et al. Diagnosing Bias and Instability in LLM Evaluation: A Scalable Pairwise Meta-Evaluator. Information 16(8):652, 2025.

[4] Chen G H et al. Humans or LLMs as the Judge? A Study on Judgement Bias. EMNLP 2024.

[5] RSNA press summary of Radiology study: Vision-based ChatGPT Shows Deficits Interpreting Radiologic Images. 2024.

[6] Oura T et al. Diagnostic accuracy of vision-language models on Japanese radiology, nuclear medicine, and interventional radiology board exams. Japanese Journal of Radiology, 2024.

[7] Vision-language model performance on the Japanese Nuclear Medicine Board Examination: high accuracy in text but challenges with image interpretation. Annals of Nuclear Medicine, 2025.

[8] Wei B. Performance Evaluation and Implications of Large Language Models in Radiology Board Exams. JMIR Medical Education, 2025.

[9] Chen L, Zaharia M, Zou J. How is ChatGPT’s behavior changing over time. arXiv:2307.09009, 2023.

[10] Yang W et al. Deep learning model inversion attacks and defenses: a comprehensive survey. Artificial Intelligence Review, 2025.

[11] Zhou Z et al. Model Inversion Attacks: A Survey of Approaches and Countermeasures. arXiv:2411.10023, 2024.

**Questions:**

1. Can you re-score a representative subset with at least one additional strong judge (open and closed), randomize candidate order, and report inter-judge agreement plus any position-bias checks and prompt/version details?

2. Can you include a stress-test on at least one expert or out-of-distribution domain, and report refusal rates and common error modes of the judge on those images?

3. Can you add one non-face MI setting and show whether FCurr inflation and the judge's corrections persist, including per-domain deltas?

---

### Official Review · Reviewer_ShTJ · 2025-11-01

**Soundness:** 3
**Presentation:** 3
**Contribution:** 2
**Rating:** 2
**Confidence:** 5

**Summary:**

This paper points out the limitation of attack accuracy, a current evaluation indicator for model inversion, and proposes using a multimodal large model to replace it.

**Strengths:**

1) They discover the equivalence between false positive in model inversion and Type I adversarial attack.
1) They manually evaluate some MLLMs, indicating that Gemini-2.0 is a reliable one.
1) They re-evaluate model inversion attacks and defenses, clarifying some long-standing misunderstandings about attack accuracy.

**Weaknesses:**

1) This paper claims to revisit model inversion evaluation, yet in reality, it only revisits attack accuracy. There are many other metrics for model inversion evaluation. In addition to the KNN distance, FID, and Knowledge Extraction Score mentioned in their related work, there are two feature distances, $\delta_{eval}$ and $\delta_{face}$ [1]. This paper lacks discussion of these metrics.
1) The core method is using a multimodal large model to evaluate attack accuracy with a simple prompt, which lacks novelty and appears more like a technical report.
1) Model inversion attack aims to reveal the information in the training data. If the reconstructed samples contain transferable features, then the privacy information has been leaked and the attack can be considered effective. For example, in their Figure 1, the reconstructed samples reveal the true skin color and eye shape. It is not necessary for them to be the same individual.

Minor revisions:
1) In Line 387, a $D_{pub}$ should be $D_{pri}$
1) $F_\text{MLLM}$ is sometimes written as $F_{MLLM}$, and $F_\text{Curr}$ is sometimes written as $F_{Curr}$

[1] A Closer Look at GAN Priors: Exploiting Intermediate Features for Enhanced Model Inversion Attacks

**Questions:**

1) What do "64" and "112" mean in "FaceNet64" and "FaceNet112"?
1) How is the correlation between $F_\text{MLLM}$ and $F_\text{Curr}$? Could you illustrate it using Pearson coefficient or a scatter plot?

---

### Official Review · Reviewer_veK4 · 2025-11-01

**Soundness:** 3
**Presentation:** 3
**Contribution:** 3
**Rating:** 4
**Confidence:** 5

**Summary:**

This paper focuses on the evaluation framework of Model Inversion Attacks (MIA). Specifically, the paper replaces the traditional evaluator with state-of-the-art Multimodal Large Language Models (MLLMs) and build a MLLM-annotated dataset for MI attack samples. Based on the new dataset, the paper points out there are significant false positives in current MI attack samples and analyzes that most of them are actually Type-I adversarial examples. Experiments results serve as evidence to support the analyses.

**Strengths:**

- Well Structured: The paper is well-structured and easy to follow.
- Interesting Insights: The paper is the first to investigate the MI evaluation framework and proposes some new insights.
- Generality of MLLMs: MLLMs do not depend on task-specific training, making it applicable to diverse MI scenarios.

**Weaknesses:**

- The study merely focuses on the white-box MI attacks. More evaluation on the black-box attacks and label-only attacks are expected.
- The experiments mainly focus on facial recognition datasets. As a benchmark, the paper should also validate whether the impact of Type-I adversarial features generalize to other non-facial domains.
- The paper lacks a comparison of the latest methods, such as [1] and [2].

[1] Peng X, Han B, Liu F, et al. Pseudo-private data guided model inversion attacks[J]. Advances in Neural Information Processing Systems, 2024, 37: 33338-33375.

[2] Li Z, Zhang H, Wang J, et al. From Head to Tail: Efficient Black-box Model Inversion Attack via Long-tailed Learning[C]//Proceedings of the Computer Vision and Pattern Recognition Conference. 2025: 29288-29298.

**Questions:**

See weaknesses

---

### Note · Authors · 2025-11-14

I have read and agree with the venue's withdrawal policy on behalf of myself and my co-authors.